# communications
# engineering

# A computational program for automated surgical planning of fenestrated endovascular repair

Tom M. Dillon [1], Patric Liang[2], Marc L. Schermerhorn [2] & Ellen T. Roche [1,3]✉

An Abdominal Aortic Aneurysm (AAA) is a dilation of the aorta at the level of the abdomen. To reduce the risk of rupture, an endograft is often implanted inside the aneurysm to decrease pressure on the aneurysm sac. To maintain blood flow to major abdominal vessels, a fenestrated endograft can be used, whereby physicians modify commercial endografts by creating fenestrations based on preoperative computed tomography imaging. The manual process of aligning patient-specific visceral anatomy onto endografts can be tedious and subject to human error. Here we developed a computational program, 'FenFit', for automated fitting of fenestrations onto commercially available endografts. A pilot clinical study was conducted to evaluate the efficiency of FenFit compared to physician manual planning, showing FenFit can reduce planning time by 62-fold on average. Our program has potential to improve clinical outcomes by providing a user interface that is expeditious and far less susceptible to human error.

[1] Department of Mechanical Engineering, Massachusetts Institute of Technology, 77 Massachusetts Avenue, Cambridge, MA 02139, USA. [2] Department of Surgery, Division of Vascular and Endovascular Surgery, Beth Israel Deaconess Medical Center and Harvard Medical School, 110 Francis St, Suite 5B, Boston, MA 02215, USA. [3] Institute for Medical Engineering and Science, Massachusetts Institute of Technology, 77 Massachusetts Avenue, Cambridge, MA 02139, USA. ✉email: etr@mit.edu

An Abdominal Aortic Aneurysm (AAA) is a dilation of the aorta, the largest blood vessel in the body, at the level of the abdomen (Fig. 1a). The incidence of AAA is 5-10 cases per 10,000 people in the U.S.[1]. Risk factors for the condition include older age, male sex, smoking, family history of AAA, hypertension, atherosclerosis, connective tissue diseases (e.g. Ehler Danos and Marfan Syndrome), and traumatic injuries to the aorta[2]. The preferred minimally invasive surgical treatment for AAA is EndoVascular Aneurysm Repair (EVAR) to reduce the likelihood of rupture[3]. During the procedure, a device is percutaneously introduced through the femoral artery, which contains a compliant tubular graft material reinforced with a metal stent mesh (an endograft).

Current EVAR devices are designed to treat AAAs that are located below the renal arteries (infrarenal AAAs). In cases where the AAA is located proximally to the visceral vessels or for aneurysms that extend across major abdominal vessels (juxtarenal, suprarenal, or thoracoabdominal aortic aneurysms), there is insufficient proximal sealing length of healthy aorta above the aneurysm to deploy a standard EVAR graft. In these cases, fenestrated endovascular repair (FEVAR) can be performed (Fig. 1b). A fenestration is a hole made in a graft to maintain blood flow to various arteries that branch from the aorta and supply blood to vital organs in the body. Following graft deployment, the fenestrations maintain blood flow to prevent end-organ ischemia[4]. Accurate fenestration placement is associated with higher branch artery patency (i.e. vessel openness), fewer postoperative endoleaks, and lower perioperative morbidity[5].

Widespread adoption of FEVAR and use in urgent cases is hindered in part by preoperative case planning challenges[6]. Fenestrated designs are inherently patient-specific and therefore cannot be easily mass-manufactured. Currently, only two medical device companies (Cook Medical, Terumo Aortic) provide a service to manufacture a patient-specific, hand-made, custom fenestrated grafts devices[7]. However, the lead time for this process can be on the order of a few weeks[8], which is often not feasible for a patient who presents with a condition requiring urgent surgical intervention. The service is also detached from the surgical planning procedure, in that there is little scope for the physician to fine-tune the design or preview the final product before it is shipped and manufactured.

Because of this lead-time and lack of accessibility to custom devices, many surgeons create their own patient-specific fenestrations on standard tube endografts[9], as summarized in Fig. 1c: (1) the surgeon uses computed tomography (CT) imaging to determine the longitudinal and circumferential positions of each fenestration relative to the graft of choice; (2) they then adjust the fenestrations using trial and error calculations, ensuring no overlap between the stent frame and fenestrations; (3) finally, while the patient is being prepared in the operating room (OR), a cautery tool is used to cut fenestrations into a sterile non-

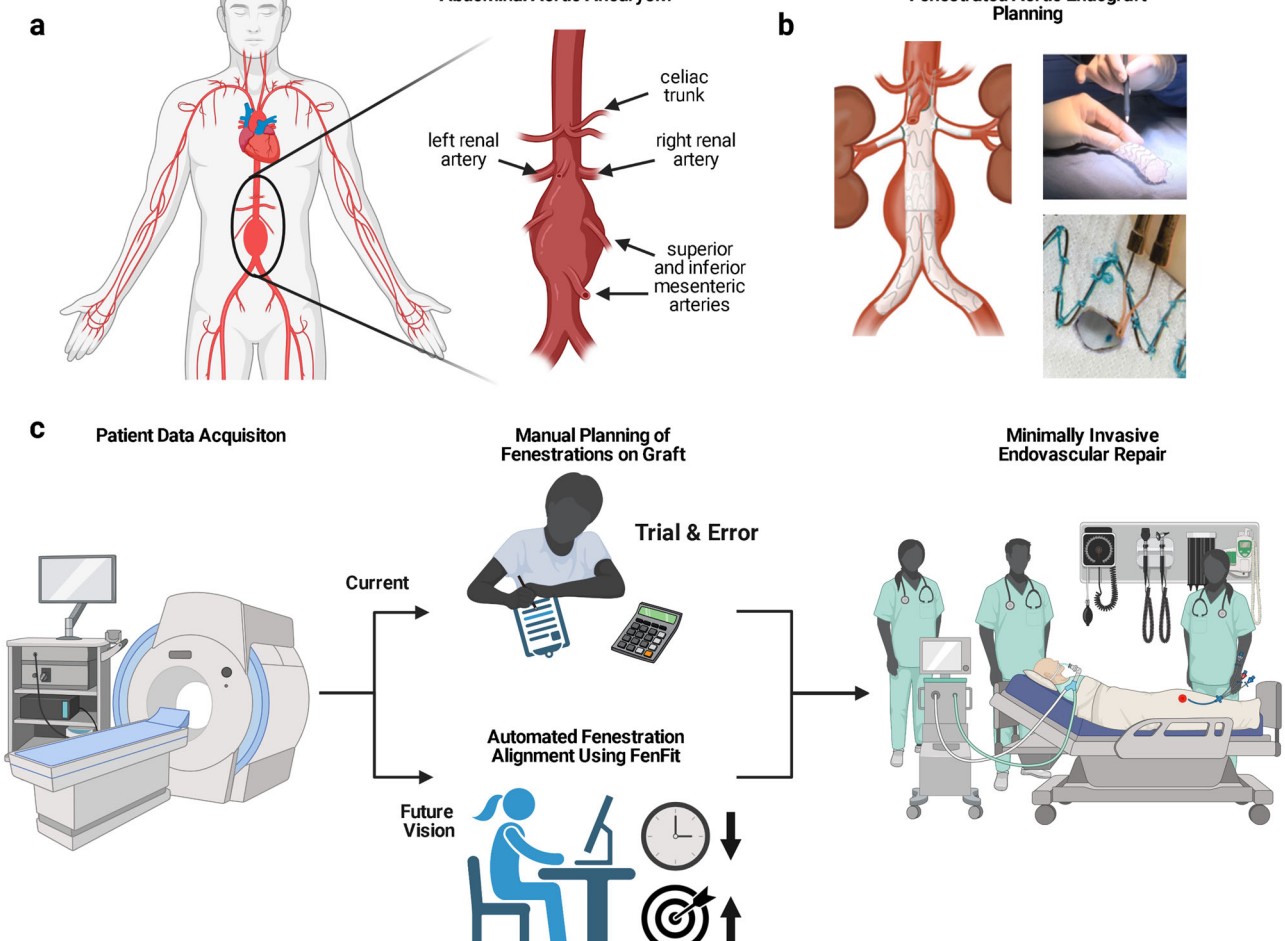

**Fig. 1 'FenFit' clinical motivation. a** An abdominal aortic aneurysm with relevant arterial vasculature highlighted. **b** A fenestrated (FEVAR) endograft, and manual modification of a commercially available EVAR graft template using a cautery tool. **c** Physician-Modified Endograft Planning (PMEG) workflow, where we propose use of our computational program – 'FenFit' – to replace manual fenestration planning.

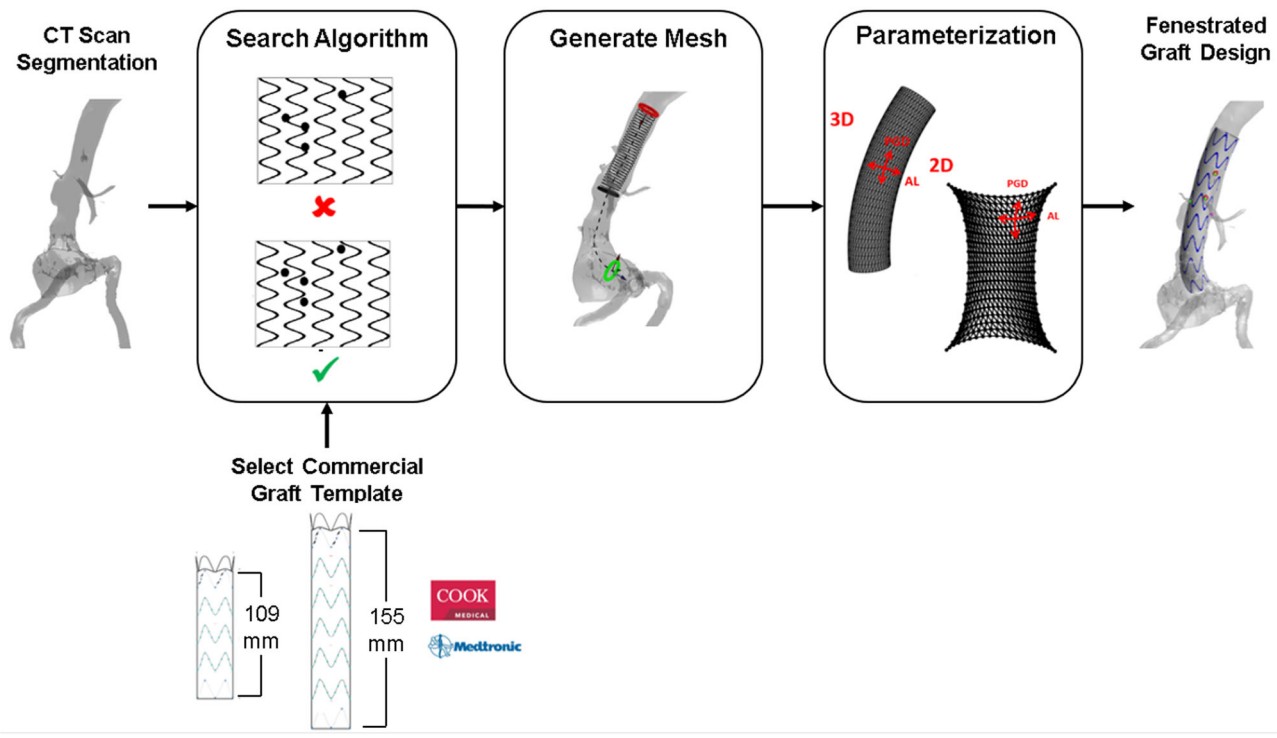

**Fig. 2 Overview of the FenFit algorithm.** The primary inputs are an AAA CT scan and a commercial graft template selected by the user. The search algorithm determines a valid alignment of the patient's anatomy given the constraints imposed by the user's selected graft design. A 3D mesh is generated inside the aorta based on the dimensions of the commercial graft template. A mesh parameterization algorithm is used to obtain a flattened 2D representation of the 3D graft mesh suitable for texture mapping. The output from FenFit is a patient-specific fenestrated endograft design. CT Computed Tomography, AL Arclength, PGD Proximal Graft Distance.

fenestrated commercially available endograft. Step 2 is the most tedious and is subject to calculation inaccuracies. For a particular graft orientation, there is no way to know a priori if fenestrations will overlap the stent struts, which cannot be cauterized. This uncertainty mandates extensive trial and error on the physician's part. There is an unmet clinical need to automate the conversion of mass manufactured grafts to patient-specific, fenestrated grafts. In the interest of improving both the speed and accuracy of surgical planning, we developed a computational program for automating the search for valid fenestration alignments on a graft, herein referred to as 'FenFit'. The following functional requirements were defined for the program: (1) maximize the accuracy and speed of the fenestration alignment process (currently the process takes ~30 min[10]), (2) allow the physician to customize any graft from a flexible design repository of both tapered and uniform diameter commercially-available graft templates, (3) visualize the final graft design in a 3D reconstructed aorta based on the preoperative CT scan, and (4) provide standardized instructions to the physician on how to modify the commercially available endograft to obtain a patient-specific fenestrated design. FenFit provides the physician with an intuitive, visual user interface to expedite the procedure for modifying endovascular grafts. The accessibility and ease of use of the program should allow a greater number of surgeons and interventionists to safely perform high-risk aortic procedures. Moreover, FenFit holds potential in the fabrication of Custom-Made-Devices (CMDs), whereby the output from the program could be fed to a numerically controlled subtractive manufacturing device.

## Results

**Design and realization of the FenFit algorithm.** FenFit was developed using MathWorks MATLAB r2020b (MathWorks, Natick, MA, USA). The primary input to the program is a segmented CT scan (obtained using 3D Slicer software), and the primary outputs are (1) a set of instructions for graft modification to be implemented in the OR, and (2) a 3D visualization of the final graft design inside the preoperative CT scan.

The primary steps involved in the FenFit algorithm are summarized in Fig. 2. To maximize computational efficiency, FenFit conducts its search algorithm in a projected 2D space, rather than the complex 3D space of the graft and patient anatomy. During the search, two image masks are generated to discretize the search space into a 2D array of pixels. The graft geometry is projected to 2D, yielding a flattened representation of the 3D geometry (*graft* mask - Fig. 3a). The fenestrations are also projected from the CT scan to a separate 2D image space (*fenestration* mask-Fig. 3b). The key dimensions measured from the CT scan are (1) the arclength (AL), or circumferential distance of a given fenestrations around the graft, and (2) the proximal graft distance (PGD), which is the distance between the fenestration and the top of the graft.

The challenge lies in determining an optimal longitudinal and rotational orientation of the fenestrations for a given graft design. This process corresponds to convolution of the image masks in the AL and PGD directions respectively. Figure 4a shows the graft and fenestration masks that are fed to the fenestration alignment search algorithm. In Fig. 4b, the fenestration mask incrementally slides over the graft mask, and valid fits are those where no overlap occurs between the fenestrations and stent struts.

After locating an optimal fit, mesh parameterization and texture mapping algorithms are used to project the fenestrations back to the graft surface in 3D and visualize the final design alongside the CT segmentation. A cylindrical graft mesh generated from the aortic centreline as well as a 2D graft alignment obtained from the search algorithm are fed to the mesh parameterization algorithm (Fig. 5a). The discrete Laplacian

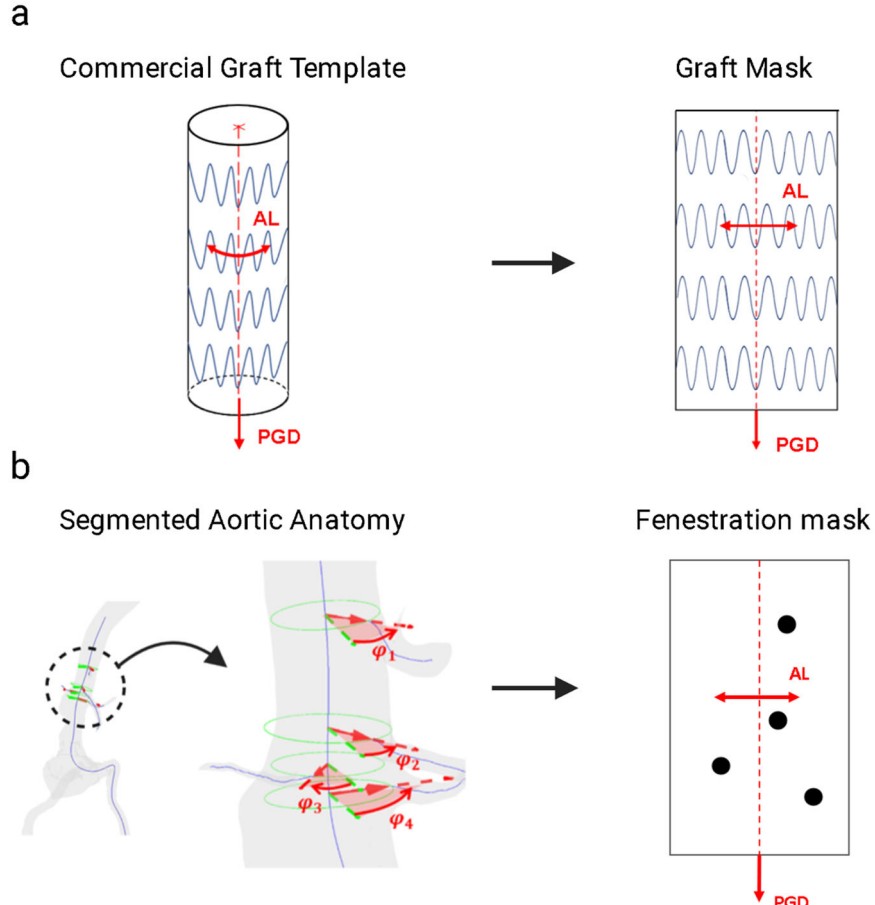

**Fig. 3 Generation of FenFit 2D image masks. a** Flattening of the 3D graft geometry in FenFit to obtain the graft mask. Uniform diameter grafts are unwound to a single rectangular search space. **b** Measurement of aortic CT scan to obtain the fenestration mask. The angles $\varphi_i$ of the fenestrations relative to an arbitrary datum vector are measured from the CT scan, facilitating 2D representation of the abdominal vasculature in the AL-PGD space. CS Coordinate System, AL Arclength, PGD Proximal Graft Distance.

approach calculates a bijective mapping between the 2D image space and 3D graft mesh providing information about the connectivity and adjacency of the graft mesh vertices (Fig. 5b). The parameterized mesh facilitates texture mapping of the optimal graft design to the mesh surface (Fig. 5c) (see *Materials & Methods* for more info).

**Experimental design**. FenFit was deployed for use in clinical trials using the MATLAB Application Compiler r2020b (Math-Works, Natick, MA, USA). The Application Compiler allows clinicians to use FenFit's user interface without any prior use of MATLAB. A single-institution retrospective review of anonymized abdominal computed tomography (CT) scans in all consecutive patients scheduled to undergo fenestrated aortic endovascular repair at Beth Israel Deaconess Medical Centre (BIDMC) from August 2020 to January 2021 was performed to evaluate the efficiency and accuracy of FenFit compared to physician manual planning. This study was approved by the BIDMC Institutional Review Board, with permission to use imaging data without the need for individual patient informed consent, due to the de-identified nature of the data. The cases were part of a submission for a Food and Drug Administration Investigational Device Exemption (FDA-IDE, clinical trial registration number: NCT04746677). The primary outcomes surgeons are concerned with in determining an optimal fit on a Cook Medical Alpha endograft are (1) the speed at which the fenestrations are aligned on the graft (quantitatively assessed via the *planning time*

referenced below), and (2) the alignment accuracy of the fenestrations relative to their corresponding arterial ostia (illustrated via the alignment *deviation*).

Pre-operative CT images were initially reviewed for treatment candidacy using the ConserusTM Enterprise Viewing System (Change Healthcare, Nashville, TN), and the determination that the patient was high risk for open aortic surgery and would be treated using a fenestrated device was determined by the treating physician. During this study period, FenFit was utilized in 25 cases. Of these FenFit cases, 14 (56%) were planned using Cook Alpha thoracic tapered grafts, and 11 (44%) using Cook Alpha thoracic straight grafts. The physician obtained the centreline of the aorta using an image reconstruction program (TeraRecon Inc., Durham, NC) and measured the initial *AL* and *PGD* for the fenestration mask manually relative to a fixed point along the aorta (typically the celiac artery when present, otherwise the superior mesenteric artery (SMA)). Although the functionality was not utilized by the physician in this study, it is worth noting that this step may be automated by FenFit using the process highlighted in Fig. 2b.

For our study, the *planning time* was defined as the time elapsed for the program or physician to find a valid fit on the aortic graft (i.e. the time to locate a valid configuration of the fenestration positions on the graft given a set of *AL* and *PGD* measurements). The *alignment deviation*, $\delta$, is defined as the distance between a graft fenestration (determined by either FenFit or the physician) and its corresponding arterial ostium (where ground truth is obtained from the segmented CT scan). $\delta$ is the

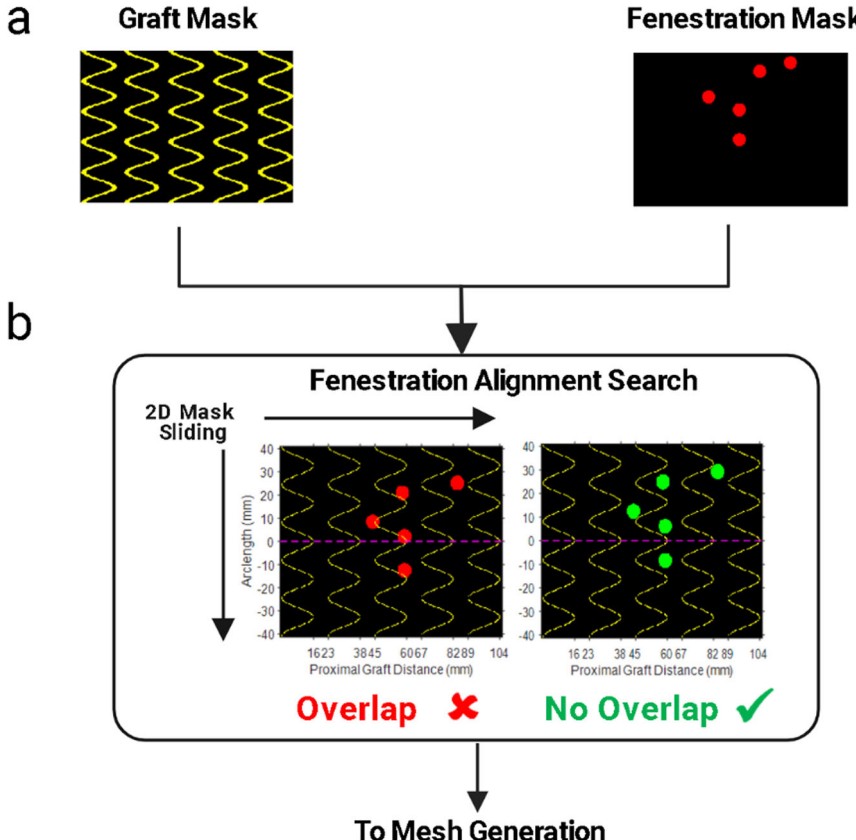

**Fig. 4 FenFit search algorithm.** Convolution of the fenestration and graft mask is performed to locate graft alignments where no overlap exists between the fenestrations and stent struts. **a** Graft and fenestration mask inputs. **b** During the search, the fenestrations slide in the arclength and proximal graft distance directions. An invalid fit is visualized in the algorithm in red, whereas valid fits are highlighted in green.

curved distance on the surface of the graft; that is, $\delta = \sqrt{\triangle AL^2 + \triangle PGD^2}$, where $\triangle AL$ and $\triangle PGD$ represent deviations in the arclength and proximal graft distance directions respectively. Planning times were recorded by the surgeon for 25 retrospective CT scans using the automated FenFit approach and traditional physician manual planning. FenFit planning and manual planning were run independently using the same fenestration AL and PGD data from the patient CT scan as input, akin to the parallel workflows illustrated in Fig. 1c. Six vascular fellows of various aortic planning experience, ranging from 0 to 3 years took part in the study. A single fellow was randomly assigned to each of the clinical cases, performed the calculations, and found the fenestration fits independently. The FenFit algorithm also selected an optimal graft design independent from the fit manually located by the physician. All plans were reviewed and finalized by the attending surgeon, MS, and adjustments made as deemed necessary. This additional time of attending review was not tracked accurately and was therefore not including in the plan time comparison. Visceral anatomy was segmented for 10 CT scans from the 25-patient cohort, allowing the alignment deviation to be calculated for those cases. The remaining 15 scans were not taken at the appropriate level of contrast necessary for fenestration segmentation, and therefore only the planning time (and not the alignment deviation) was measured in these cases. All data was reviewed by attending surgeons experienced in complex aortic case planning at BIDMC. The two primary outputs from FenFit are graft modification instructions on where to cauterize the fenestrations on the endograft in the OR, as well as a 3D overlay of the final fenestrated graft design alongside the aortic segmentation. In addition to these outputs, the user is given an estimate for the placement accuracy of the FenFit program.

Figure 6 depicts the primary outputs as well as the alignment deviation for each of the visceral fenestrations for 3 patients from the study cohort. proposes the graft alignment The FenFit algorithm seeks to minimizes the deviation, $\delta_f$, between the ground truth location of the patient's vessel (obtained from the CT scan) and the fenestration centers mapped from 2D to 3D, as depicted in Fig. 6a. The corresponding 2D graft alignment to be implemented in the operating room is shown in Fig. 6b. The physician alignment deviation, $\delta_m$ (where the fenestration centre is estimated via manual inspection of CT scan slices), is depicted on the same plot for comparison of program accuracy.

Figure 7a shows the primary outputs for 6 patients from the study cohort, illustrating the range of anatomies that FenFit was applied to. The fenestrated graft design is seen to conform well to the curvature of the aorta for even the most tortuous of cases, owing to the mesh parameterization and texture mapping algorithm implemented (described in *Material and Methods*). Note that grafts are typically supported by an infrarenal segment at their distal end (shown with dashed lines).

As desired, the fenestration alignment program resulted in a significantly shorter fenestration planning time compared to surgeon planning as illustrated in Fig. 7b.

For the 10-patient cohort with segmentable anatomy, Fig. 7c demonstrates that FenFit can achieve vessel deviations in the submillimeter range. It is evident from both the qualitative results presented in Fig. 6, as well as the quantitative evaluation of deviation illustrated in Fig. 7c, that FenFit yields a much lower alignment deviation on average. Alignment deviation in the

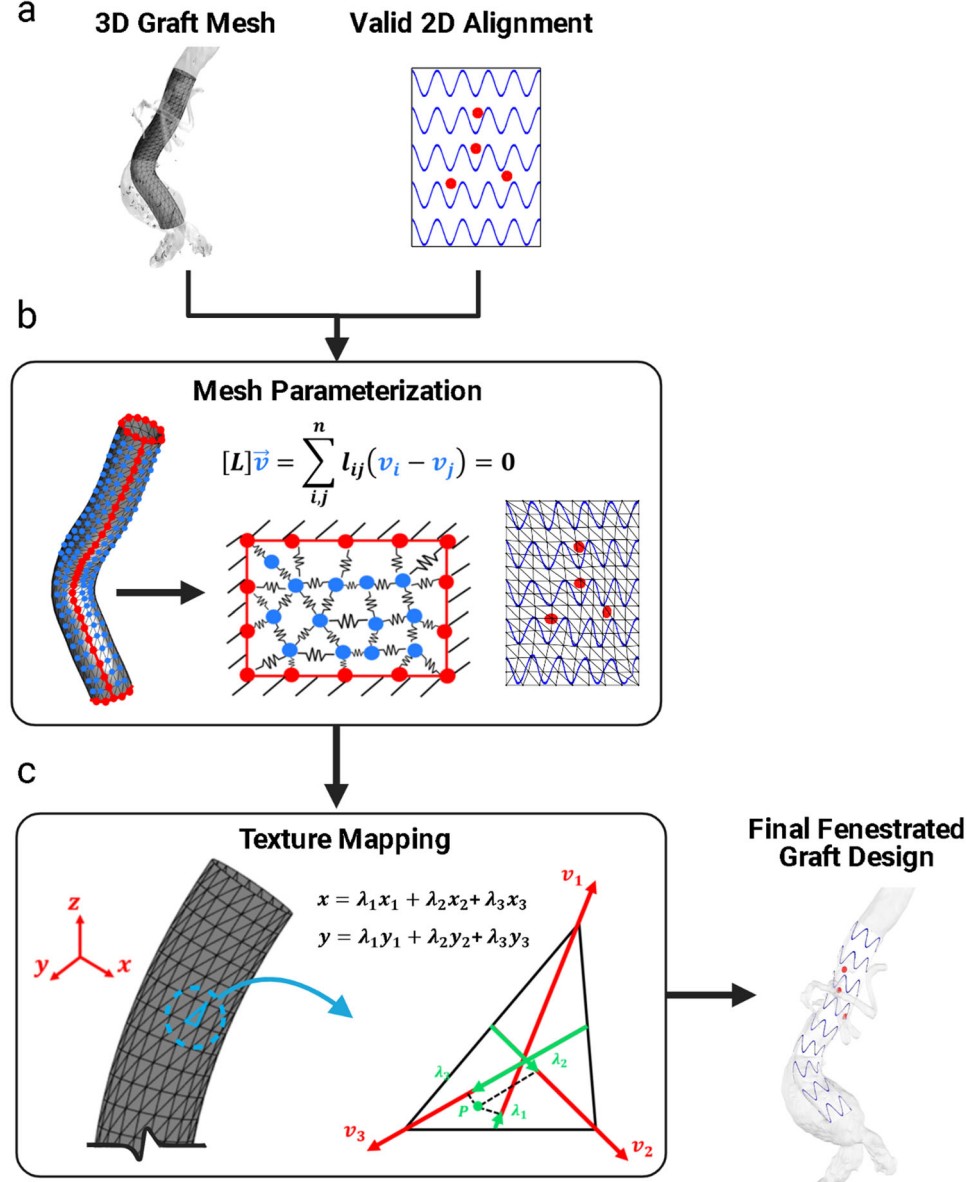

**Fig. 5 Mesh Parameterization algorithm in FenFit. a** A 3D graft mesh and alignment are available following Steps 1 and 2 in FenFit. **b** Mesh parameterization involves dimensionality reduction techniques used to obtain a 2D representation of a 3D geometry. The Laplacian matrix $L$ (with entries $l_{ij}$) encapsulates the geometric relationship between an arbitrary set of vertices $v_i$ and $v_j$ in the 3D domain. **c** Texture mapping was used to interpolate discrete portions of the image between the mesh nodes. Any 2D position $(x, y)$ bounded within a triangular mesh element with vertices $(x_1, y_1)$, $(x_2, y_2)$, and $(x_3, y_3)$ can be represented in 3D by barycentric coordinates $\lambda_1$, $\lambda_2$, and $\lambda_3$.

manual planning case can be attributed to human error, whereas discrepancies from FenFit arise either from discretization errors that accumulate during the mapping process, or from vessel prioritization, whereby the fenestration is allowed to adjust independently by a small amount to obtain a valid alignment (see Supplementary Fig. S4, Supplementary Methods). 2 cases from the 25-patient cohort required non-prioritized vessels to obtain a valid fit (see Supplementary Table S.1 for more information). If no fit is possible based on re-orientation of the graft alone, 3 mm of posterior deviation in the arclength direction is permitted for the renal fenestrations (based on the MS and PL's experience with successful cannulation of the renal arteries with this amount of posterior displacement).

Demographics of the patients and anatomic details of the automated planning group are listed in Supplementary Table S1 in the Supplementary Methods section. Both FenFit and manual

alignment deviations are provided in Supplementary Table S2, Supplementary Methods.

## Discussion

This work introduces an automated program for patient-specific fenestration alignment along endovascular grafts. A search algorithm was developed that determines a valid alignment between the patient's anatomy and a commercial graft template selected by the user, and a mesh parameterization algorithm was developed to obtain a flattened 2D representation of the 3D graft mesh suitable for texture mapping.

Our program addresses an unmet clinical need by automating the conversion of mass manufactured grafts to patient-specific fenestrated grafts, increasing both the speed and accuracy of surgical planning. Our retrospective clinical study has shown that

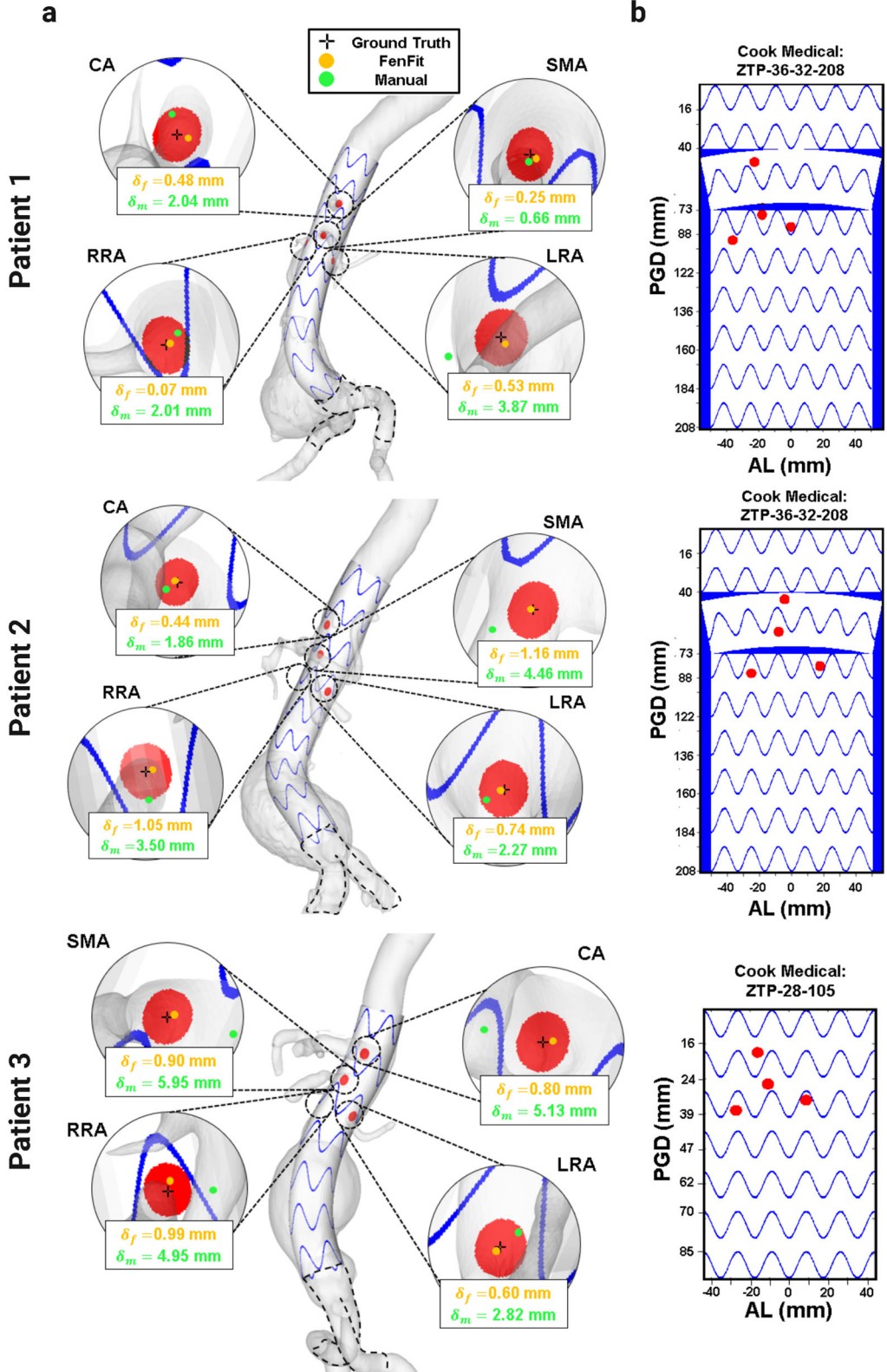

**Fig. 6 FenFit primary outputs. a** 3D fenestrated graft visualization, including alignment deviation. The crosshairs represent the ground truth location of the patient's vessels (based on CT scan), the orange markers ($\delta_f$) represent the optimal graft fenestration centers mapped by FenFit, and the green markers ($\delta_m$) represent the manually measured ostia position by the surgeon. **b** 2D graft fenestration instructions to be implemented in the operating room. SMA superior mesenteric artery, CA celiac artery, RRA right renal artery, LRA left renal artery.

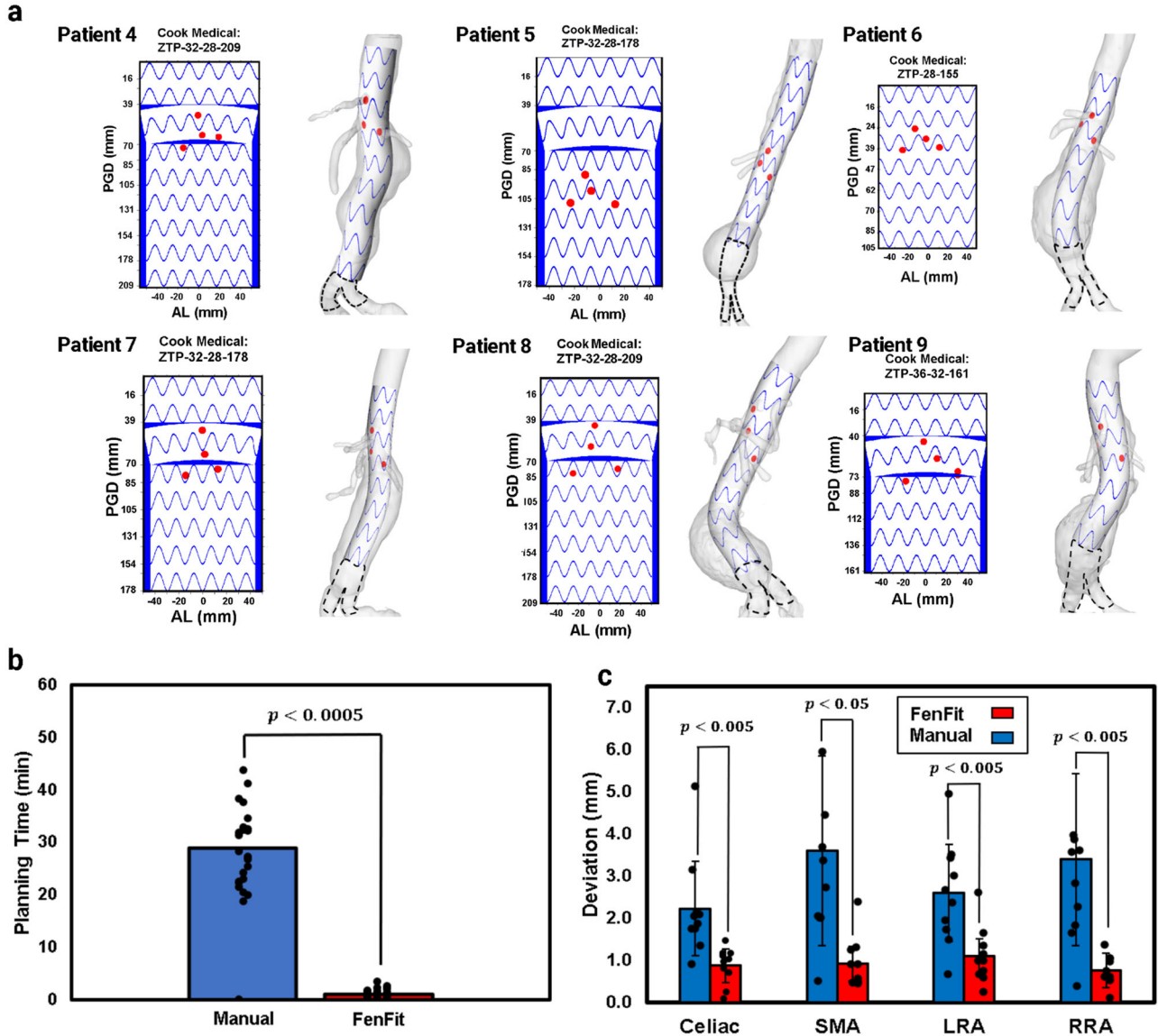

**Fig. 7 Summary of clinical study results. a** OR instructions and 3D graft visualization for 6 patients from the 10-patient cohort. **b** fenestration planning time for manual physician and automated FenFit approach for entire study cohort (Student's $t$ test, $p < 0.0005$, $n = 25$). **c** Average deviation between each fenestration and its corresponding ostia using FenFit vs. manual planning, conducted for all cases where visceral segmentation was possible from the CT scan (normal based 95% CI: the error bar is defined as 'mean ± standard deviation', $n = 10$). SMA superior mesenteric artery, CA celiac artery, RRA right renal artery, LRA left renal artery.

FenFit can reduce fenestration planning time from 30 min to 32 s per case on average. This reduced labor time implies substantial cost savings to hospitals and institutions, facilitating extra time that vascular surgeons can spend on other patient needs. For urgent cases that require immediate medical attention, FenFit could allow surgeons to plan and intervene much faster.

It is important to note that the predicted deviation presented in Fig. 7c is not a clinical result, and instead represents the misalignment between the FenFit mapped fenestration positions and the ground truth ostia locations as per the preoperative CT scan. Deviations that arise during implantation of the graft may be larger due to aortic deformation, as well as misalignment of the graft by the surgeon. A prospective clinical study is currently underway, where FenFit has been used on 40 cases to date. In future, we intend to conduct clinical validation of our program's accuracy by comparing the fenestration positions measured from postoperative CT scans to results predicted by FenFit. Moreover, it should be noted that only a single vascular fellow was assigned

for manual planning of each case. In our future work, we will include multiple vascular surgeons per case, to better understand the variability in case planning times that may arise amongst physicians.

We estimate our planning deviation to be in the sub-millimeter range, minimizing errors that may arise during the graft modification process. Accurate planning simplifies the implantation procedure by facilitating easier branch artery cannulation through precise alignment of fenestrations with branch artery ostia[11]. Ease of cannulation can further lead to more branch vessels being addressed, shorter procedure and fluoroscopy times, lower contrast material use, and lower rates of type I or III endoleaks (postoperative graft leakage due to poor limb attachment) as well as branch vessel occlusion.

Of note, other intraprocedural factors can also affect the difficulty of successful target branch vessel cannulation. Once the fenestrated graft is inserted into the aorta, the graft is partially deployed and rotated until the correct orientation is achieved and

aligned with the branch vessels. Graft fenestrations are made with radiopaque markers so that their orientation can be visualized on fluoroscopy[12]. This alignment process is facilitated by intraoperative angiography, pre-stented branch vessels, or 3D imaging overlay mapping using prior obtained computed tomography angiography images, if this technology is available at the institution. Tortuous aortic anatomy and small diameter access vessels can make graft rotation and graft-anatomy alignment accuracy more challenging.

Visualization of the final fenestrated graft alongside the CT scan should allow physicians to gain trust in the efficacy of FenFit to produce a valid design. We designed the algorithm to be flexible to a wide range of graft designs and aortic anatomies, allowing the algorithm to perform robustly for even the most tortuous cases. We anticipate that our program will ultimately translate to improved clinical outcomes, by providing an expedited intuitive user interface (UI) that is far less susceptible to human error than manual planning. The four stage UI for FenFit is highlighted in Supplementary Fig. S6, Supplementary Methods, and Supplementary Video SV 1. Further, we envision that the accessibility and ease of use of the program will allow a greater number of surgeons and interventionists to safely perform these high-risk surgical procedures.

Further work may be required to incentivize surgeons to utilize our program and expedite translation to the clinic. To fully automate the planning workflow from raw CT scan to fenestrated graft design, a method for automated segmentation would be necessary. For our study, aortic segmentations were obtained using semiautomatic segmentation tools available in most medical imaging software (e.g. TeraRecon Inc.), though many surgeons may not be equipped to use these tools. Manual operations required within these software packages include selection of an appropriate threshold to isolate the aortic anatomy, use of seed growth tools for identifying the aortic lumen, and smoothing operations on the final anatomy. In future, we may explore a fully integrated environment in FenFit for automated aortic segmentation, fenestration planning, and graft visualization. Moreover, our program requires the surgeon to manually select a commercial graft template prior to fenestration fitting. It may be better for FenFit to iterate over multiple graft templates and recommend a design to the surgeon based on additional clinical parameters such as the amount of proximal or distal seal, or graft tortuosity.

It is worth noting that extensive work has been done using finite-element modeling to predict tissue deformation that arises following implantation of fenestrated stent grafts[13-15]. Avril et al.[16] provide a good review of predictive modeling of EVAR in clinical applications, including virtual stent graft deployment simulation based on CT preoperative imaging, and post-EVAR blood flow dynamics[13,15-17]. One such study of interventional modeling is authored by Derycke et al.[18]. Simulation was used to predict the post-operative positions of the fenestrations and graft sizing following EVAR deployment. The model was based on finite element analysis and assessed the deformations induced by the device-host interaction yielding a prediction of arterial displacement. Moreover, recent literature has highlighted advances in using computational tools to aid the design of EVAR devices deployed in virtual models. Hemmler et al.[19] developed an in-silico model of customized stent grafts that have the same morphology as the underlying luminal vessel surface for improved EVAR implantation with reduced likelihood of postoperative complications such as endoleaks and graft migration. While these studies seek to gain a better understanding of the mechanics of graft deployment that determine post-operative shape, none are implemented as a device-specific, automated design tool for minimizing trial and error graft planning. To our knowledge, this is the first automated search algorithm that accounts for the patient-specific constraints imposed by both the stent struts and visceral anatomy.

The FenFit algorithm could be expanded to treat almost all aneurysm types (thoracic, femoral, carotid etc.). For instance, our mesh parameterization algorithm is well suited to conduct the alignment process in the more tortuous anatomy of the aortic arch. FenFit may also have applications beyond cylindrical endovascular grafts alone – for example, bioprosthetic valves used in transcatheter valve repair may require fenestrations to maintain the patency of the left and right coronary arteries.

Augmented reality registration of FenFit results to an intraoperative angiogram has the potential to reduce intervention time as well as relieve cognitive effort on the surgeon's part. In future studies, we may consider intraoperative registration of the fenestrated graft design to X-ray imaging, to provide an ideal configuration for surgeons to target during intervention, further standardizing the surgical workflow.

Finally, to automate the manufacture of fenestrated grafts, the output of FenFit could be fed to a subtractive manufacturing device (e.g. a laser cutter) for precise, automated modification of commercial endograft templates. Measurement, design, and fabrication could all be completed within just a few minutes, freeing up hospital resources, while reducing preoperative planning time and costs.

## Methods

**Image Mask Generation (Step 1).** The primary input to the program is a segmentation of the aortic anatomy obtained from the CT scan. Segmentation and automated extraction of the aortic centreline was conducted using 3D Slicer software. It is worth noting that the aortic anatomy is typically segmented by a specialist prior to EVAR intervention for the purposes intraoperative mapping; therefore, segmentation would not be considered an extraneous step in the FenFit workflow required for the program to work. In future versions, we will create an integrated environment for both semi-automated segmentation and patient-specific fenestration alignment, albeit semi-automated segmentation tools are well-established in the literature[20] and are not the focus of the present study.

A parametric design repository of commercially available graft designs is used, such that only a few key variables of the graft (i.e. graft length, strut size, number of strut rings) are necessary to introduce additional graft masks in FenFit. Tapered grafts may also be added to the repository, as illustrated in Supplementary Fig. S1, Supplementary Methods, provided additional design parameters are specified for obtaining a flattened representation (Supplementary Fig. S2, Supplementary Methods). To reduce computational expense, the fenestration mask and graft mask are divided into pixels. These discrete components can be easily manipulated in FenFit using MATLAB's Image Processing Toolbox. The relative distances between the fenestrations are summarized in 2D by the fenestration mask, and the primary geometric parameters of the graft template are represented by the graft mask. For more information on the method utilized to project the 3D CT scan fenestrations to 2D fenestration mask, see Supplementary Fig. S3, Supplementary Methods.

**Search Algorithm (Step 2).** The purpose of the search algorithm is to determine valid alignments of the patient's fenestrations on the selected graft template. The primary goal is to search for graft configurations where no overlap exists between the stent struts and the fenestrations. During the search process, the relative distance between the fenestrations remains fixed and the fenestration mask incrementally slides over the graft mask in the *AL* and *PGD* directions (corresponds to rotational and axial movement of the graft along the aorta). In rarer circumstances where a valid fit cannot be found, a revised search strategy is employed by FenFit (see Supplementary Fig. S4, Supplementary Methods).

**Graft Mesh Generation (Step 3).** The third step in FenFit is to generate a 3D cylindrical mesh that conforms to the aortic segmentation, representing the deployed shape of the graft. The aortic centreline is first determined by calculating the centroid of each aortic cross section and smoothing this data to obtain a spline of best fit. To ensure the graft conforms to the aortic wall (while providing an interference fit to prevent migration of the graft), the deployed graft diameter is calculated based on the diameter of the aorta's proximal landing zone (defined as the first 10% of the graft's length as measured from the top of the graft). Finally, a lofting function from the MATLAB GIBBON visualization toolbox[21] was used to sweep a cylindrical graft mesh along the previously determined spline of best fit. The final mesh can also be varied based on whether a uniform or tapered graft is selected by the surgeon (see Supplementary Fig. S5, Supplementary Methods, for more details)

**Mesh Parameterization Algorithm (Step 4).** To render the search algorithm results on the surface of the graft mesh, a mapping function (i.e., mesh parameterization) is required between the pixels in the 2D *PGD-AL* space and the mesh

vertices in the 3D. The mesh parameterization needs to account for local curvature of the graft's surface inside the aorta.

Many mesh parameterization algorithms utilize a discrete Laplacian matrix calculation to represent the connectivity, spacing, and angles between the mesh vertices in 2D[22–24]. The discrete Laplacian $L$ is a superposition of 2 matrices that capture key geometric characteristics about the 3D mesh,

$$L = D - W \qquad (1)$$

where $D$ is the degree matrix that describes the number of nodes a given node $i$ is connected to, and $W$ is the adjacency matrix, which is a binary matrix that describes whether an arbitrary pair of nodes $i$ and $j$ are connected via an edge.

A Graph embedding algorithm was employed which utilizes a linear system of equations involving the Laplacian matrix to solve for the locations of the nodal coordinates in 2D. Specifically, we wish to obtain solutions for $v$ in the linear system,

$$Lv = b \rightarrow \sum_{i,j}^{n} l_{ij}\left(v_i - v_j\right) = b \qquad (2)$$

where $v$ is a column vector representing the positions of the nodal coordinates in the 2D domain, and $b$ is a column vector of boundary conditions, representing whether a given node $i$ is a border node ('1') or internal node ('0'). The subscript $ij$ represents an edge connecting the nodes $i$ and $j$. The external shape of the 2D nodal coordinates was constrained to a rectangle via appropriate definition of the border nodes, rendering the parameterization compatible with the results obtained from the 2D search algorithm. Border nodes and internal nodes are highlighted in red and blue in Fig. 5 respectively.

iIf the standard discrete Laplacian matrix is implemented as described above, all off-diagonal entries in $W$ will take on uniform binary values, yielding a uniformly spaced rectangular grid in 2D when Eq. (2) is solved (i.e., a trivial solution). Therefore, information about connectivity alone does not preserve the geometric properties of the 3D mesh. The relative distances between the nodes in 2D should be stretched to account for local variations in curvature of the graft. To provide a more realistic mapping, a common variation of graph embedding uses the cotangent weight Laplacian, first proposed by Pinkall and Polthier[25]. In this representation, the relative distances between the nodes in the 3D domain is used to modify the entries in the adjacency matrix $W$. The modified Laplacian entries are calculated by,

$$l_{ij} = \begin{cases} w_{ij} = \frac{1}{2}(cot\,\alpha_{ij} + cot\,\beta_{ij}) & \text{if } ij \text{ is an edge} \\ -\sum_{k}^{N} w_{ik} & \text{if } i = j, \text{otherwise } w_{ik} = 0 \end{cases} \qquad (3)$$

where the angles $\alpha_{ij}$ and $\beta_{ij}$ are the angles opposite to the edge $ij$ and $N$ represents the number of elements in a single row of the Laplacian matrix.

Cotangent functions are utilized given they provide a measure of the aspect ratio of a mesh element. Therefore, the absolute distance between the nodes can vary between the 3D and 2D domains while still preserving the angles between the nodes. Overall, the cotangent Laplacian matrix can be used to generate a deformed 2D representation of the graft mesh.

**Texture Mapping (Step 5)**. The final step in FenFit is to conduct texture mapping of the 2D graft alignment (obtained in step 2) to the 3D graft mesh (step 3) via the parameterized mesh obtained using the Laplacian (step 4). Given a 1-1 mapping exists between the 2D and 3D nodes, the objective of the texture mapping algorithm is to accurately interpolate discrete portions of the image between the nodes. A barycentric coordinate system was selected to specify arbitrary locations on the graft's surface (not necessarily the nodal coordinates). In this representation, the location of any point bounded within a triangular element on the graft's surface can be expressed as a sum of weights or "masses" at each of its vertices[26], as illustrated in Fig. 5. Therefore, barycentric coordinates can provide a 3D coordinate system for mapping individual pixels from the mesh parameterization to the graft.

We can write the Cartesian coordinates of a point $r$ on the graft's surface as a function of both the barycentric coordinates and the local triangular vertices,

$$x = \lambda_1 x_1 + \lambda_2 x_2 + \lambda_3 x_3 \qquad (4)$$

$$y = \lambda_1 y_1 + \lambda_2 y_2 + \lambda_3 y_3 \qquad (5)$$

where $x$ and $y$ define a point bounded within the triangular vertices $(x_1, y_1)$, $(x_2, y_2)$, and $(x_3, y_3)$, and $\lambda_1$, $\lambda_2$, and $\lambda_3$ are the barycentric coordinates that sum to $\lambda_1 + \lambda_2 + \lambda_3 = 1$. Note that locally on the surface of the graft, we can assume the mesh is flat and approximately 2 dimensional.

We require an inverse mapping of Eqs. (4) and (5); that is, a transformation that converts $(x, y)$ coordinates on the 2D image to barycentric coordinates suitable for rendering in the 3D graft space. To obtain this transformation, we first rearrange the identity $\lambda_1 + \lambda_2 + \lambda_3 = 1$ and substitute into Eqs. (4) and (5) above,

$$x - x_3 = \lambda_1(x_1 - x_3) + \lambda_2(x_2 - x_3)$$

$$y - y_3 = \lambda_1(y_1 - y_3) + \lambda_2(y_2 - y_3)$$

Or in matrix form,

$$(r - r_3) = T\lambda$$

where,

$$T = \left[(x_1 - x_3)(x_2 - x_3)(y_1 - y_3)(y_2 - y_3)\right]$$

Finally, to obtain the desired mapping, we invert the matrix $T$,

$$\lambda = T^{-1}(r - r_3) \qquad (6)$$

where, explicitly, the components of $T^{-1}$ can be calculated by,

$$\lambda_1 = \frac{(y_2 - y_3)(x - x_3) - (x_3 - x_2)(y - y_3)}{\det(T)}$$

$$\lambda_2 = \frac{(y_3 - y_1)(x - x_3) - (x_1 - x_3)(y - y_3)}{\det(T)}$$

To implement the equations above in practice, each triangular mesh face was initially discretized into barycentric coordinates using Eq. (6). For each face on the deformed 2D mesh, its corresponding 3D vertices were extracted, and a 1-1 mapping was assigned. Finally, the image was interpolated across the graft's surface, according to each pixel's mapping under the barycentric coordinate system.

**Statistical analysis**. Microsoft excel was used for statistical analysis and graphs. For comparison of manual and automated planning times, a Student's $t$ test was performed on all available cases ($N = 25$). Results were considered significant when a $P$ value of ≤0.05 was obtained. Error bars indicate 1 standard deviation in all plots.

**Reporting summary**. Further information on research design is available in the Nature Portfolio Reporting Summary linked to this article.

## Data availability

We have made the following data publicly available on *figshare*[27]:
- De-identified patient CT segmentations fed as input to the FenFit program.
- Beth Israel Deaconness Medical Center Boston (BIDMC) FenFit Study Protocol
- Raw excel file used by surgeons at Beth Israel Deaconess Medical Centre, Boston which was used to compile Figs. 7b and 7c, as well as tables S7 and S2.
- A repository of 2D graft modification instructions output from the FenFit algorithm (as illustrated in Figs. 6 and 7).

The FenFit algorithm and 3D visualizations of the final graft design will be available upon reasonable request from the authors.

## Code availability

Given the authors have filed a patent application related to this work (U.S. Patent application number PCT/US2021/060591), the custom code implemented in this manuscript is not available to the public at the time of writing. In the event of patent publication, the code will be available upon reasonable request from the authors.

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

## Acknowledgements

The authors have filed a patent application (U.S. Patent application number PCT/US2021/060591). Clinical trial registration number: NCT04746677, Beth Israel Deaconess Medical Center, Boston. This project was funded with the generous support of the MIT MathWorks School of Engineering Fellowship.

## Author contributions

Algorithm development – T.D. Clinical Testing & Feedback – P.L., M.S. Supervision – E.T.R., P.L., M.S. Writing – original draft: T.D. Writing – review and editing: T.D., P.L., E.T.R., M.S.

## Competing interests

The authors declare no competing interests.
