## [Peer Review File · Communications Engineering]

Title: A Computational Program for Automated Surgical Planning of Fenestrated Endovascular RepairReviewers' comments:

Reviewer #1 (Remarks to the Author):

In this paper, the authors propose a computational program for automated surgical planning of fenestrated endovascular repair. The application is the creation of fenestrations on off-the-shelf endografts just before a procedure.

Generally, when this can be anticipated, suppliers such as Cook Medical or Terumo Aortic can supply stent grafts with personalized fenestrations or branches.

When a fenestrated stent graft is needed in emergency and only off the shelf stent grafts are available, the computer program FenFit proposed by the authors can be very useful as it provides a mask in less than a minute to cut precisely the fenestrations before the procedure.

The approach is very interesting and the manuscript is well presented.

There remains a couple of points to clarify:

Major :

1. the approach was applied retrospectively on 25 patients but the deviations between the positions of fenestrations and the actual positions of ostia is only estimated for 10. It should be explained why the others were excluded.

2. If I understand well, the estimation of deviations in Fig 7C is not a clinical result. This is the prediction of deviations by the computer program, but the actual deviations when the stent graft is implanted can be larger due to deformations of the graft, deformations of the aorta and mispositioning by the practitioner. It should be clearly acknowledged that this predicted deviation, done on a retrospective cohort, is not a clinical result. The real validation will come with the prospective study and the analysis of post-operative scans, which is underway as the authors mention the use of FenFit on 12 patients prospectively in the discussion (results are not provided).

3. there are some reference missing on the topic of surgical planning of fenestrated endovascular repair. The authors mention Cook Medical but not Terumo Aortic as suppliers of fenestrated stent grafts. Moreover, extensive work was done using finite-element modeling to automate the design of fenestrated stent grafts, see Derycke et al, Eur J Vasc Endovasc Surg, vol 59, issue 2 ? pp 237-246 (2020). Check also this recent review : <https://onlinelibrary.wiley.com/doi/pdf/10.1002/cnm.3529>.

Typo :

Line 210, it is instead of is it

In ref 6, Journal instead of Journa

Reviewer #3 (Remarks to the Author):

This manuscript describes the characteristics of 'FenFit', a software for automatic graft fenestration planning. In the paper, FenFit has been applied to abdominal aortic aneurysm (AAA) planning. The application is interesting and a needed step in planning, as it replaces manual planning which might take some time and skills.

Please consider the following comments:

1. My understanding is that the paper describes how to make the fenestrations in the graft in a way that is consistent and optimal (i.e. using FenFit). However, when placing the graft the fenestrations will need to correctly align with the anatomy. This is not considered in this paper, but it will be nice to mention how this is done, and whether the surgeon has the ability to place the graft and move it around until good alignment occurs.

2. Lines 198-200 describe the criteria for a 'good fit'. How is this quantified? Using the software (FenFit)? Or independently by a clinician - from some measurements? Please clarify. An independent measure would be best.

3. Lines 217-220 describe how the study comparisons were done on data from AAA patients. Were the 6 fellows assigned to first do a manual fit (time themselves) and then use FenFit (and time themselves)? Was the order random? And were the 6 fellows assigned to all the 25 cases? And from there, how was the alignment accuracy quantified?
4. Is FenFit fenestration algorithm completely automatic? Could the fellows be inputting data biased by the manual fitting? Please comment on the manuscript as well.
5. Figure 7 describes the results. Figure 7B, however, differs from the data presented in Supplementary Table S1. Likely the table needs to be updated, as it seems that perhaps manual times and FenFit times were inverted (at least according to the results shown in Figure 7B). Why are there no errors in table S1 reported? (didn't you have different fellows doing the same AAA to account for user differences?). If not, this will be an important consideration that either needs to be added to the manuscript or at least acknowledge as a limitation of the current study.
6. Figure 7B seems to consider deviations from FenFit - but how about the comparison to the manual fit? What were those?
7. Line 286, quickly describes implantation and alignment. Please comment on how alignment is done during implantation.
8. Lines 317-320. What is this? Future work? It is not clear.

The authors of manuscript (No. COMMSENG-22-0262-T) are very thankful to reviewers for their comments to further improve the quality of this work. We have addressed the comments raised by the reviewers and the detailed response for each comment is provided below.

Reviewer 1 Comments

In this paper, the authors propose a computational program for automated surgical planning of fenestrated endovascular repair. The application is the creation of fenestrations on off-the-shelf endografts just before a procedure. Generally, when this can be anticipated, suppliers such as Cook Medical or Terumo Aortic can supply stent grafts with personalized fenestrations or branches. When a fenestrated stent graft is needed in emergency and only off the shelf stent grafts are available, the computer program FenFit proposed by the authors can be very useful as it provides a mask in less than a minute to cut precisely the fenestrations before the procedure.

The approach is very interesting and the manuscript is well presented. There remains a couple of points to clarify:

Comment #1 The approach was applied retrospectively on 25 patients but the deviations between the positions of fenestrations and the actual positions of ostia is only estimated for 10. It should be explained why the others were excluded.

Response #1 We thank the reviewer for the suggestion to clearly distinguish between the 25 patient and 10 patient cohort for our readership. On lines 223-226 of the manuscript, we introduce the 10-patient cohort and explain why the deviation was not calculated for the other 15 cases:

“Visceral anatomy was segmented for 10 CT scans from the 25-patient cohort, allowing the alignment deviation to be calculated for each case. The remaining 15 scans were not taken at the appropriate level of contrast necessary for fenestration segmentation, and therefore only the planning time (and not the alignment deviation) was measured in these cases.”

Where the expression in parentheses above has been added for further clarity.

The 10-patient cohort is only relevant for Figure 7C in this study. Therefore, later in the manuscript when Figure 7C is first introduced, we add the following phrase on lines 255-257:

“For the 10-patient cohort with segmentable anatomy, Figure 7C demonstrates that FenFit can achieve vessel deviations in the sub-millimeter range.”

Comment #2 If I understand well, the estimation of deviations in Fig 7C is not a clinical result. This is the prediction of deviations by the computer program, but the actual deviations when the stent graft is implanted can be larger due to deformations of the graft, deformations of the aorta and mispositioning by the practitioner. It should be clearly acknowledged that this predicted deviation, done on a retrospective cohort, is not a clinical result. The real validation will come with the prospective study and the analysis of post-operative scans, which is underway as the authors mention the use of FenFit on 12 patients prospectively in the discussion (results are not provided).

Response #2 We thank the reviewer for emphasizing the importance of acknowledging limitations of our prospective study. A more extensive explanation of study limitations has been added to lines 283-284 of the manuscript. We reiterate the definition of alignment deviation so that the reader does not confuse

this result with implantation accuracy. We also briefly mention our ongoing work where true clinical validation of the FenFit program will be sought:

“...For urgent cases that require immediate medical attention, FenFit could allow surgeons to plan and intervene much faster. It is important to note that the predicted deviation presented in Figure 7C is not a clinical result, and instead represents the misalignment between the FenFit mapped fenestration positions and the ground truth ostia locations as per the preoperative CT scan. Deviations that arise during implantation of the graft may be larger due to aortic deformation, as well as misalignment of the graft by the surgeon. A prospective clinical study is currently underway, where FenFit has been used on 40 cases to date. In future, we intend to conduct clinical validation of our program’s accuracy by comparing the fenestration positions measured from postoperative CT scans to results predicted by FenFit.”

Comment #3 there are some references missing on the topic of surgical planning of fenestrated endovascular repair. The authors mention Cook Medical but not Terumo Aortic as suppliers of fenestrated stent grafts. Moreover, extensive work was done using finite-element modeling to automate the design of fenestrated stent grafts, see Derycke et al, Eur J Vasc Endovasc Surg, vol 59, issue 2 ? pp 237-246 (2020). Check also this recent review: <https://onlinelibrary.wiley.com/doi/pdf/10.1002/cnm.3529>.

Response #3

The reviewer brings up an important point regarding elucidation as to where FenFit fits in the context of existing literature on finite element modeling for fenestrated stent grafts. The manuscript has been updated with an additional paragraph (following line 310) that discusses finite element modeling for endovascular repair, while also making the distinction that FenFit’s contribution is to offer an automated and expedited design tool for interventional planning. Drawing from the review paper mentioned in the comment above, we discuss a range of EVAR simulation papers; spanning deformation modeling, hemodynamic analyses, interventional assessment, and (most adjacent to our work) recent development in computational tools to aid graft design.

“It is worth noting that extensive work has been done using finite-element modelling to predict tissue deformation that arises following implantation of fenestrated stent grafts [12], [13], [14]. Avril et al. [15] provide a good review of predictive modelling of EVAR in clinical applications, including virtual stent graft deployment simulation based on CT preoperative imaging, and post-EVAR blood flow dynamics [14], [16]. One such study of interventional modeling is authored by Derycke et al. [17]. Simulation was used to predict the post-operative positions of the fenestrations and graft sizing following EVAR deployment. The model was based on finite element analysis and assessed the deformations induced by the device-host interaction yielding a prediction of arterial displacement. Moreover, recent literature has highlighted advances in using computational tools to aid the design of EVAR devices deployed in virtual models. Hemmler et al. [18] developed an in-silico model of customized stent grafts that have the same morphology as the underlying luminal vessel surface for improved EVAR implantation with reduced likelihood of postoperative complications such as endoleaks and graft migration. While these studies seek to gain a better understanding of the mechanics of graft deployment that determine post-operative shape, none are implemented as a device-specific, automated design tool for minimizing trial and error graft planning. To our knowledge, this is the first automated search algorithm that accounts for the patient-specific constraints imposed by both the stent struts and visceral anatomy.”

Terumo Aortic are now also mentioned as additional vendors of fenestrated stent grafts (on line 82).

Comment #4 Line 210 it is instead of is it; In ref 6, Journal instead of Journa – both corrected

Reviewer 3 Comments

This manuscript describes the characteristics of 'FenFit', a software for automatic graft fenestration planning. In the paper, FenFit has been applied to abdominal aortic aneurysm (AAA) planning. The application is interesting and a needed step in planning, as it replaces manual planning which might take some time and skills.

Comment #1 My understanding is that the paper describes how to make the fenestrations in the graft in a way that is consistent and optimal (i.e. using FenFit). However, when placing the graft the fenestrations will need to correctly align with the anatomy. This is not considered in this paper, but it will be nice to mention how this is done, and whether the surgeon has the ability to place the graft and move it around until good alignment occurs.

Response #1 To address the reviewer's comment, surgeons and co-authors of this paper P.L. and M.S. have provided a more in-depth discussion of the graft alignment procedure, as well as the factors that determine intraoperative alignment accuracy. This information has been added after line 290 in our manuscript.

“...Ease of cannulation can further lead to more branch vessels being addressed, shorter procedure and fluoroscopy times, lower contrast material use, and lower rates of type I or III endoleaks (postoperative graft leakage due to poor limb attachment) as well as branch vessel occlusion. *Of note, other intraprocedural factors can also affect the difficulty of successful target branch vessel cannulation. Once the fenestrated graft is inserted into the aorta, the graft is partially deployed and rotated until the correct orientation is achieved and aligned with the branch vessels. Graft fenestrations are made with radiopaque markers so that their orientation can be visualized on fluoroscopy. This alignment process is facilitated by intraoperative angiography, pre-stented branch vessels, or 3D imaging overlay mapping using prior obtained computed tomography angiography images, if this technology is available at the institution. Tortuous aortic anatomy and small diameter access vessels can make graft rotation and graft-anatomy alignment accuracy more challenging.*”

Comment #2 Lines 198-200 describe the criteria for a 'good fit'. How is this quantified? Using the software (FenFit)? Or independently by a clinician - from some measurements? Please clarify. An independent measure would be best.

Response #2 We thank the reviewer for identifying the need for providing further details on what constitutes an optimal fit for our program. As discussed in the manuscript, the physician cares most about obtaining an accurate fit as quickly as possible. This calls for the definition of two quantitative metrics to assess the performance of FenFit highlighted in the 'Experimental Design' section of our manuscript – namely, the alignment deviation and planning time.

To make the connection clearer between qualitative user needs and the quantitative metrics presented in Figures 7B and 7C, we have added the following phrases in parentheses to our manuscript:

“...The cases were part of a submission for a Food and Drug Administration Investigational Device Exemption (FDA-IDE, clinical trial registration number: NCT04746677). *The primary outcomes the physician is concerned with in determining an optimal fit on a Cook Medical Alpha endograft are (1) the speed at which the fenestrations are aligned on the graft (quantitatively assessed via the planning time*

referenced below), and (2) the alignment accuracy of the fenestrations relative to their corresponding arterial ostia (illustrated via the alignment deviation).”

The physician independently measures the planning time for both study cases using a stopwatch (FenFit vs. manual). The alignment deviation is obtained using the formula $\delta = \sqrt{AL^2 + PGD^2}$, where AL and PGD are the deviations in the arclength and proximal graft distance, calculated retrospectively for both the FenFit and physician planning processes. This information is presented in lines 212-216:

“For our study, the planning time was defined as the time elapsed for the program or physician to find a valid fit on the aortic graft (i.e. the time to locate a valid configuration of the fenestration positions on the graft given a set of AL and PGD measurements). The alignment deviation, δ , is defined as the distance between graft fenestration (determined by FenFit or the physician) and its corresponding arterial ostium (where ground truth was obtained from the segmented CT scan). Here, δ is the curved distance on the surface of the graft; that is, $\delta = \sqrt{\Delta AL^2 + \Delta PGD^2}$, where ΔAL and ΔPGD represent deviations in the arclength and proximal graft distance directions respectively. Planning times were recorded by the surgeon for the 25 retrospective CT scans using the automated FenFit approach and traditional physician manual planning”

The last statement was added to make it clear that measurements are taken *after* both the planning algorithm and manual process are complete (to highlight these are independent measures from the design search algorithm itself).

Comment #3 Lines 217-220 describe how the study comparisons were done on data from AAA patients. Were the 6 fellows assigned to first do a manual fit (time themselves) and then use FenFit (and time themselves)? Was the order random? And were the 6 fellows assigned to all the 25 cases? And from there, how was the alignment accuracy quantified?

Response #3 We thank the reviewer for bringing up this clarification. To address this, we have updated lines 218-220 of the manuscript:

“Six vascular fellows of various aortic planning experience, ranging from 0 to 3 years, took part in the study. A single fellow was randomly assigned to each of the clinical cases, and this fellow performed the calculations, and found the fenestration fits independently.”

We understand that by not cross-assigning surgeons to FenFit cases, the study may fail to capture variability associated with the surgeon’s interventional planning experience. Therefore, we add this as a limitation of the study (following lines 284 in the original manuscript) and hope to address potential variability in our ongoing prospective clinical work:

“In future, we intend to conduct clinical validation of our program’s accuracy by comparing the fenestration positions measured from postoperative CT scans to results predicted by FenFit. Moreover, it should be noted that only a single vascular fellow was assigned for manual planning of each case. In our future work, we will include multiple vascular surgeons per case, to better understand the variability in case planning times that may arise amongst physicians.”

As mentioned in response #2 above, alignment accuracy was calculated using the formula $\delta = \sqrt{\Delta AL^2 + \Delta PGD^2}$.

Comment #4 Is FenFit fenestration algorithm completely automatic? Could the fellows be inputting data biased by the manual fitting? Please comment on the manuscript as well

Response #4 Figure 2 in the manuscript illustrates the automated workflow of FenFit. As detailed in this figure, FenFit assumes that a CT segmentation of the graft is available for measuring visceral anatomy, and that the surgeon specifies a graft design template they would like to modify to produce a FEVAR graft. The output offers a set of instructions on where to cauterize the graft in the operating room, as well as a 3D visualization of the final graft for verification purposes.

The program does not, however, automate the segmentation process for obtaining the initial 3D geometry of the aneurysmal anatomy. This is highlighted as a limitation of our work (lines 303-307). Some of the semi-automatic steps involved with obtaining this 3D geometry have been added to the manuscript, to paint a clearer picture of manual tasks that remain in the workflow:

“To fully automate the planning workflow from raw CT scan to fenestrated graft design, a method for automated segmentation would be necessary. For our study, aortic segmentations were obtained using semi-automatic segmentation tools available in most medical imaging software (e.g. TeraRecon Inc.), though many surgeons may not be equipped to use these tools. Manual operations required within these software packages include selection of an appropriate threshold to isolate the aortic anatomy, use of seed growth tools for identifying the aortic lumen, and smoothing operations on the final anatomy. In future, we may explore a fully integrated environment in FenFit for automated aortic segmentation, fenestration planning, and graft visualization.”

Nevertheless, FenFit drastically reduces planning time via implementation of the search algorithm detailed in the “design and realization of FenFit” section.

Regarding the potential for bias the reviewer mentions, there is no way for the results of the manual fitting process to influence the FenFit algorithm and vice versa. As depicted in Figure 1C, both processes were run in parallel and independently for the study using only the same input data from the CT scan. In the algorithm design section, it is mentioned that FenFit searches over all possible configurations of the graft and selects the best one. Our algorithm makes its selection independent of the results obtained from the manual fitting process. To make this point clear for the reader, we have added the following to the ‘Experimental design’ section (line 218):

“Planning times were recorded by the surgeon for the 25 retrospective CT scans using the automated FenFit approach and traditional physician manual planning. FenFit planning and manual planning were run independently using the same fenestration AL and PGD data from the patient CT scan as input, akin to the parallel workflows illustrated in Figure 1C. Six vascular fellows of various aortic planning experience, ranging from 0 to 3 years, took part in the study. A single fellow was randomly assigned to each of the clinical cases, performed the calculations, and found the fenestration fits independently. The FenFit algorithm also selected an optimal graft design independent from the fit manually located by the physician.”

Comment #5 Figure 7 describes the results. Figure 7B, however, differs from the data presented in Supplementary Table S1. Likely the table needs to be updated, as it seems that perhaps manual times and FenFit times were inverted (at least according to the results shown in Figure 7B). Why are there no errors in table S1 reported? (didn't you have different fellows doing the same AAA to account for user differences?). If not, this will be an important consideration that either needs to be added to the manuscript or at least acknowledged as a limitation of the current study.

Response #5 Supplementary Table S1 was indeed inverted and has been corrected now. We thank the reviewer for pointing this out. As per response #3 above, only a single surgeon was assigned to each case which has been acknowledged as a limitation of the study.

Comment #6 Figure 7B seems to consider deviations from FenFit - but how about the comparison to the manual fit? What were those?

Response #6 We thank the reviewer for highlighting a key comparison metric that was initially omitted from our work. Comparisons of alignment deviation are now included for both FenFit and manual planning to better illustrate the improved accuracy FenFit can offer.

A sample patient from Figure 6 is shown below. Rather than showing only the FenFit location and ground truth ostia location as per the CT scan as before, we now include an additional location - that of the physician identified ostia location obtained via inspection of CT scan slices. Both FenFit and the physician attempt to keep the fenestrations as close to the ground truth as possible (now represented by a set of crosshairs, so the reader understands this is a target for both). Deviations for FenFit and manual fitting are also indicated by δ_f and δ_m respectively.

We update the reader's introduction to this figure in the results section:

“Figure 6 depicts the primary outputs as well as the alignment deviation for each of the visceral fenestrations for 3 patients from the study cohort. FenFit proposes the graft alignment that minimizes the deviation, δ_f , between the ground truth location of the patient's vessel (obtained from the CT scan) and the fenestration centres mapped from 2D to 3D. The physician alignment deviation, δ_m (where the fenestration centre is estimated via manual inspection of CT scan slices), is depicted on the same plot for comparison of program accuracy.”

Figure 7C provides a summary of the deviation alignment results. Alignment deviations for both FenFit and manual physician planning are now provided (shown below and updated in the manuscript):

“Figure 7 – Summary of clinical study results.... (c) Average deviation between each fenestration and its corresponding ostia using FenFit vs. manual planning, conducted for all cases where visceral segmentation was possible from the CT scan (normal based 95% CI: error bars show mean \pm standard deviation, n=10). SMA, superior mesenteric artery; CA celiac artery; RRA right renal artery; LRA left renal artery.”

Figure 7C is now discussed in the results section as follows:

“For the 10-patient cohort with segmentable anatomy mentioned above, Figure 7C demonstrates that FenFit can achieve vessel deviations in the sub-millimeter range. It is evident from both the qualitative results presented in Figure 6 as well as the quantitative evaluation of deviation illustrated in Figure 7C that the alignment deviation is much lower on average in the case of FenFit. Alignment deviation in the manual planning case can be attributed to human error, whereas discrepancies from FenFit planning can arise either from discretization errors that accumulate during the mapping process, or from vessel prioritization...”

Comment #7 Line 286 quickly describes implantation and alignment. Please comment on how alignment is done during implantation.

Response #7 See response #1 above

Comment #8 Lines 317-320. What is this? Future work? It is not clear.

Response #8 Lines 317-320 have been updated to clarify FenFit registration as a potential future work:

“Augmented reality registration of FenFit results to an intraoperative angiogram has the potential to reduce intervention time as well as relieve cognitive effort on the surgeon’s part. In future studies, we may consider intraoperative registration of the fenestrated graft design to X-ray imaging to provide an ideal configuration for surgeons to target during intervention, further standardizing the surgical workflow.”

REVIEWERS' COMMENTS:

Reviewer #1 (Remarks to the Author):

thanks for addressing all my comments

Reviewer #3 (Remarks to the Author):

The authors have addressed all concerns from previous reviews. I have no further concerns.